# OccDriver: Future Occupancy Guided Dual-branch Trajectory Planner in Autonomous Driving

**Zhao Huang**[1*]   **Bowen Zhang**[2]   **Zhongzhu Li**[2]   **Di Lin**[1†]
[1]Tianjin University    [2]SenseAuto

## Abstract

Trajectory planning for autonomous driving is challenging due to agents' behavioral uncertainty and intricate multi-agent interaction modeling. Most existing studies generate trajectories without explicitly exploiting possible scene evolution, while world models predict consequences from ego behavior, enabling more informed planning decisions. Inspired by the world model, we propose OccDriver, a novel rasterized-to-vectorized dual-branch framework for trajectory planning. This pipeline performs a coarse-to-fine trajectory decoding process: The vectorized branch first generate multimodal coarse trajectories; Then the rasterized branch predicts future scene evolutions conditioned on each coarse trajectory via occupancy flow prediction; Lastly, the vectorized branch leverages intuitive future interaction evolution of each modality from the rasterized branch and produces refined trajectories. Several cross-modality (occupancy and trajectory) losses are further introduced to improve the consistency between trajectory and occupancy prediction. Additionally, we apply a contingency objective in occupancy space, considering marginal and joint occupancy distributions in different planning scopes. Our model is assessed on the large-scale real-world nuPlan dataset and its associated planning benchmark. Experiments show that OccDriver achieves state-of-the-art in both Non-Reactive and Reactive closed-loop performance.

## 1 Introduction

Trajectory planning for autonomous driving confronts intrinsic challenges due to the complexity of multi-agent interactions and pervasive uncertainty Djuric et al. (2020); Xu et al. (2014)—from sensor noise to high-level behavioral unpredictability. Deep learning methods Bansal et al. (2018); Guo et al. (2023); Scheel et al. (2022) have emerged as a promising alternative to rule-based systems Bouchard et al. (2022); Treiber et al. (2000); Yi et al. (2018). However, effectively modeling the dynamic and uncertain interplay among agents in future scenarios remains a formidable task. In this work, we tackle this problem from two key perspectives: predictive modeling of future interactions and the choice of its representation space.

Representation space is crucial in planning, transforming raw sensor inputs into structured features that capture environmental contexts. Figure. 1 presents a comparison of different representation paradigms. Rasterized representations Hu et al. (2021); Kim et al. (2022); Hu et al. (2023b), by modeling occupancy over spatiotemporal grids, offer robustness against occlusions and a probabilistic view of joint future states Liu et al. (2023a); Mahjourian et al. (2022). Unfortunately, this approach incurs discretization artifacts and loses fine individual context and geometric details. In contrast, vectorized methods Jiang et al. (2023); Zhou et al. (2022; 2023) provide high-precision trajectory generation by capturing detailed individual semantics but tend to oversimplify evolving future interactions and require substantial manual feature engineering to approximate uncertainty Khaitan et al. (2021); Chen et al. (2024). To overcome these limitations, we propose a hybrid approach that retains the probabilistic strengths of rasterized joint modeling while preserving the individual fidelity of vectorized representations, supporting probabilistic interaction modeling and more interaction-informed trajectory planning.

---

*Work done during internship at SenseAuto.
†Corresponding author.

Figure 1: Illustration of different representation paradigms: (a) rasterized framework models the scene-level joint dynamics in spatiotemporal grids; (b) vectorized framework performs individual-level trajectory planning; (c) our proposed dual-branch framework integrates scene-level and individual-level information, further leveraging future scene as planning guidance.

The problem of interaction modeling has been widely studied in recent years. Multi-agent reinforcement learning methods Kiran et al. (2021); Liu et al. (2022) are developed to simulate interactions and learn policies through trial and error. However, they often struggle with scalability and environmental non-stationarity. Graph neural networks (GNNs) Mo et al. (2022); Sheng et al. (2022) excel in capturing relational dependencies, yet suffer from message-passing limitations and escalating computational costs as the number of agents increases. Beyond encoding mechanisms, decoding mechanisms e.g., game theoretic approaches Huang et al. (2023); Wang et al. (2021) and tree policy planning Huang et al. (2024) have been explored. However, these methods typically lack robustness, limiting practicality for real-world deployment. Nevertheless, all these methods perform forward-only planning without correction ability when poor rollout occurs, often necessitating a strong trajectory scoring module. Our method first plans multimodal coarse ego trajectories in vectorized branch. Then the rasterized branch constructs probabilistic occupancy maps conditioned on each coarse trajectory, capturing future scene evolution resulting from each ego action. Lastly, the interactive knowledge embedded in the occupancy space is distilled into the vectorized branch for trajectory guidance.

Beyond the framework, we propose a suite of specialized loss functions, leveraging future occupancy as planning guidance. Occupancy interference loss captures ego and agents' exclusivity in occupancy space, which can be seen as ego planning in occupancy space. Occupancy guidance loss enforces consistency between trajectories and predicted occupancy, which bridges the gap between occupancy distribution and trajectory planning, ensuring effective information transfer between the two branches.

Our framework also integrates contingency planning objective Cui et al. (2021); Li et al. (2023b); Liu et al. (2024a) utilizing both marginal and joint occupancy distribution to balance safety and efficiency in dynamic traffic environments. In short-term, we estimate marginal occupancy probabilities of key interactive agents, enabling the ego vehicle to respond swiftly to imminent risks. For long-term planning, we estimate joint occupancy probabilities to construct a modality-consistent traffic evolution, ensuring scene-compliance and avoiding over-conservative behavior.

Our contributions are summarized as follows:

1) We propose a dual-branch transformer framework for coarse-to-fine trajectory planning, where a rasterized branch serves as a 2D occupancy world model by predicting future scene evolution conditioned on coarse trajectories and guiding fine-grained trajectory planning.

2) We introduce several specialized losses to facilitate effective information transfer between vectorized and rasterized branches, imposing intuitive guidance on planning.

3) We incorporate a contingency planning strategy that leverages short-term marginal occupancy distribution for risk-sensitive planning and long-term joint occupancy distribution for scene-compliant behavior generation.

4) Extensive evaluation on the NuPlan dataset, achieving state-of-the-art planning performance on both reactive and non-reactive closed-loop metrics.

## 2 RELATED WORK

### 2.1 REPRESENTATION SPACE IN MOTION PLANNING

Imitation-based planning method Hu et al. (2022a); Cheng et al. (2024b) has attracted lots of research interest due to the accessibility of massive real-world expert driving data Caesar et al. (2020; 2021);

Ettinger et al. (2021). It can be categorized into two branches according to the representation paradigm: rasterized and vectorized approaches.

**Rasterized Approaches** project scene context into discret bird-eye-view(BEV) images Hu et al. (2022b); Li et al. (2024c) and encode it with off-the-shelf image feature extraction methods Liu et al. (2021b); Dosovitskiy et al. (2020). Earlier research uses CNN Renz et al. (2022); Song et al. (2020) for feature encoding and trajectory decoding, while recent research utilizes transformer structure Chen et al. (2021); Huang et al. (2022); Chitta et al. (2022a); Zhang et al. (2022); Yao et al. (2023) for better performance. Some works decode agents' future movement as future occupancy and flow fields Liu et al. (2023a); Kim et al. (2022), providing a dense and intuitive scene representation, yet compromising fewer individual details and limited receptive field.

**Vectorized Approaches** yield impressive performance because of vector's concise but effective representation capacity for scene semantics. Building on vectorized representation of traffic scenes Gao et al. (2020), researchers have used structures like graph neural networks for interaction modeling and DETR-style transformers Wang et al. (2023); Achaji et al. (2022) for query-based decoding. Additionally, instead of fully learnable queries, anchor-based Afshar et al. (2024); Li et al. (2023a); Chen et al. (2024) queries are used to decode multi-modal trajectories with explicit patterns.

OccDriver combines the advantages of vectorized and rasterized representations. The rasterized branch predicts future occupancy and flow fields to represent future scene evolution, serving as guidance for trajectory planning. The comparison between our work and other occupancy-assisted planning methods is provided in Appendix A.2.

## 2.2 CONTINGENCY PLANNING

Contingency planning Li et al. (2023c) is traditionally framed as a tree-structured trajectory optimization problem, where each branch represents a possible scenario and a shared short-term segment ensures safety across all cases. While theoretically complete, this approach suffers from exponential complexity, as branches grow combinatorially with interactive agents' intentions, requiring a complicated safety-evaluation pruning strategy. Also, each agent's intention is reduced to a deterministic approximation, resulting in the loss of trajectory-level multi-modality.

OccDriver addresses these challenges by formulating contingency planning within a dense probabilistic occupancy space. Rather than explicitly constructing and pruning a scenario tree, it directly predicts multiple scene-level rollouts of interactive agents' joint occupancy distribution, which also mitigates the limitations of deterministic scenario approximation.

## 3 METHOD

The overall framework of OccDriver is illustrated in Fig. 2. We first introduce problem formulation in Sec .3.1. Then, we demonstrate OccDriver's dual-branch network architecture in Sec. 3.2. Marginal occupancy distribution prediction is presented in Sec. 3.3. Finally, in Sec. 3.4, we propose several specialized training losses.

## 3.1 PROBLEM FORMULATION

Our research is dedicated to the task of trajectory planning. Our model input $\mathbf{X}$ composes of states of ego vehicle $\mathbf{E}$ and dynamic agents $\mathbf{A}$ over historical horizon $T_h$, states of static objects $\mathbf{S}$ and a High-Definition map $\mathbf{M}$. The objective is to plan ego vehicle's $M$-modal future states $\mathbf{Y} = \{(y_i, \pi_i) \mid i = 1...M\}$, where $y$ is trajectory over future horizon $T_f$ and $\pi$ is the confidence score. With the integration of the occupancy branch, our model is updated with rasterized inputs and outputs. $\mathbf{E}^0$, $\mathbf{A}^0$ and $\mathbf{M}$ at current step are projected into occupancy grid $\mathbf{O}_e^0$, $\mathbf{O}_a^0$ and $\mathbf{O}_m$. Following the practice in Liu et al. (2023a), current backward flow $\mathbf{FL}^0$ is computed as extra input. Occupancy prediction branch takes $\mathbf{X}_{occ} = \{\mathbf{O}_e^0, \mathbf{O}_a^0, \mathbf{O}_m, \mathbf{FL}^0\}$ as input and predicts ego vehicle's occupancy $\mathbf{O}_e$, other agents' occupancy $\mathbf{O}_a$ and scene's backward flow $\mathbf{FL}$ over future horizon $T_f$. In alignment with trajectory planning, occupancy branch outputs multimodal prediction $\mathbf{Y}_{occ} = \{(\mathbf{O}_{e,i}, \mathbf{O}_{a,i}, \mathbf{FL}_i) \mid i = 1...M\}$. Consequently, the formulation of our model is given as:

$$\mathbf{Y}, \mathbf{Y}_{occ} = f(\mathbf{X}, \mathbf{X}_{occ} \mid \theta), \tag{1}$$

where $f$ denotes the neural network of OccDriver, $\theta$ is the model parameters.

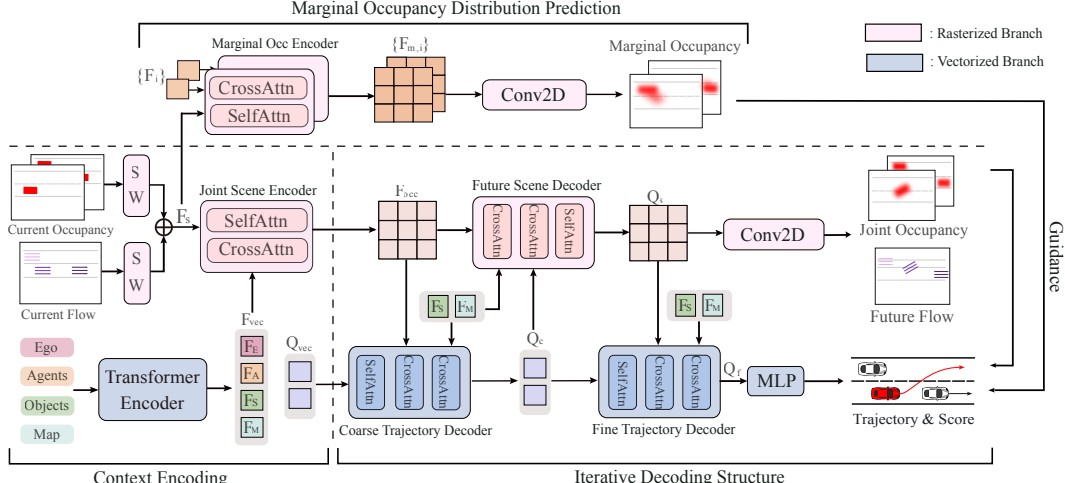

Figure 2: **The architecture of the OccDriver** comprises three fundamentals. Context Encoding first encodes heterogeneous inputs into vectorized individual features $F_{vec}$ and joint scene feature $F_{occ}$ respectively. Initialized by $Q_{vec}$ and $F_{occ}$, dual-branch iterative decoding structure decodes joint future occupancy and trajectory iteratively. Short-term marginal occupancy is generated via marginal occupancy distribution prediction. Joint and marginal occupancy predictions enforces explicit guidance to trajectory through specialized losses.

## 3.2 DUAL-BRANCH ARCHITECTURE

**Context Encoding**. For vectorized branch, heterogeneous inputs $\{\mathbf{E}, \mathbf{A}, \mathbf{S}, \mathbf{M}\}$ are encoded as individual features $\{F_E, F_A, F_S, F_M\}$ with separate encoders. After added by positional embedding, encoded features are concatenated as $F_{vec} \in \mathbb{R}^{(1+N_A+N_S+N_M) \times D}$. We then perform scene-level feature fusion via a transformer encoder, employing self-attention to capturing social interactions between encoded scene semantics.

For the rasterized occupancy branch, $\mathbf{O}^0 = \{\mathbf{O}_e^0, \mathbf{O}_a^0, \mathbf{O}_m\}$ and $\mathbf{FL}^0$ are embedded into occupancy feature $F_o$ and flow feature $F_f$ with separate Swin-Transformer blocks. $F_o$ and $F_f$ are then concatenated and projected into the scene feature map $F_s \in \mathbb{R}^{(H/4) \times (W/4) \times D}$ with an MLP. Then we propose a two-layer attention-based joint scene encoder for better aggregation of scene information. Cross-attention is employed to integrate coarse trajectory feature $F_{vec}$'s dense semantic information, followed by a self-attention for feature fusion in the occupancy space. The process of joint scene encoder is described as:

$$F_s = \text{CrossAttn}(F_s, F_{vec}, F_{vec}), \tag{2}$$

$$F_{occ} = \text{SelfAttn}(F_s), \tag{3}$$

where $\text{CrossAttn}(Q, K, V)$ and $\text{SelfAttn}(X)$ indicates cross-attention and self-attention mechanism respectively. The encoded occupancy feature is denoted as $F_{occ}$.

**Iterative Decoding Structure.** Retaining encoded vectorized feature $F_{vec}$ and occupancy feature $F_{occ}$, we decode future occupancy evolution and future trajectories iteratively. As depicted in Fig. 2, iterative decoding structure is composed of 3 sequential decoders: coarse trajectory decoder $D_c$, future scene decoder $D_s$ and fine trajectory decoder $D_f$. For better consistency in scene evolution, we jointly decode the future trajectories of ego vehicle and agents. A set of $M$ learnable queries $Q_m \in \mathbb{R}^{M \times D}$ is combined with $\{F_E, F_A\} \subset F_{vec}$, forming multi-modal decoding queries $Q_{vec} \in \mathbb{R}^{M \times (N_A+1) \times D}$.

Coarse trajectory decoder $D_c$ takes $Q_{vec}$ as input, deriving coarse trajectory queries $Q_c$. Each layer of $D_c$ comprises of three types of attention mechanisms. Within each modality, $Q_{vec}$ first performs self-attention to extract social interactions among agents. Then cross attention is employed to integrate static obstacle and map information from $\{F_S, F_M\} \subset F_{vec}$. Lastly, $Q_{vec}$ queries $F_{occ}$ via another cross-attention, establishing a spatial scene understanding.

In occupancy branch, future scene decoder $D_s$ acts as a world model in BEV view, decoding $F_{occ}$ into feature $Q_s$ that represent future scene evolution conditioned on ego coarse trajectories. In each layer of $D_s$, $F_{occ}$ conducts two cross-attention operations to query $Q_c$ and $\{F_S, F_M\}$ respectively,

aggregating instance-level features and map information. $F_{occ}$ further applies self-attention for social interaction modeling. The decoded $Q_s$ encompasses an intuitive prior of the future scene evolution.

Despite aggregating current scene information, $Q_c$ requires further refinement under the guidance of future scene evolution. Thus, utilizing future scene information from $Q_s$, fine trajectory decoder $D_f$ refines $Q_c$ into future-informed trajectory query $Q_f$ to make scene-consistent planning. $D_f$ and $D_c$ share the same architecture, except that $D_f$ leverages $Q_s$ instead of $F_{occ}$ as Key and Value in the third cross-attention. The iterative decoding process is formulated below:

$$Q_c = D_c(\mathrm{Q} = Q_{vec},\ \mathrm{K}, \mathrm{V} = F_{occ}, \{F_S, F_M\}), \tag{4}$$

$$Q_s = D_s(\mathrm{Q} = F_{occ},\ \mathrm{K}, \mathrm{V} = Q_c, \{F_S, F_M\}), \tag{5}$$

$$Q_f = D_f(\mathrm{Q} = Q_c,\ \mathrm{K}, \mathrm{V} = Q_s, \{F_S, F_M\}). \tag{6}$$

**Prediction heads.** Given $Q_f$ and $Q_s$, different prediction heads are implemented. Fine trajectory feature $Q_f$ determines $M$-modal joint planning trajectories $\mathbf{Y}$ with corresponding confidence scores $\pi$ via two MLPs. To generate reasonable raw behaviors, coarse trajectory feature $Q_c$ is decoded into coarse trajectory $\mathbf{Y_c}$ via another MLP. Future scene feature $Q_s$ is upsampled back to original shape of the input images and then deployed with two separate 2d-CNNs to output $M$-modal future occupancy fields and flow fields. Different from earlier works, we decouple the future occupancy prediction into the ego vehicle $\mathbf{O}_e$ and surrounding vehicles $\mathbf{O}_a$ to facilitate explicit loss guidance detailed in Sec. 3.4. We provide further architectural details in Appendix. C.

## 3.3 MARGINAL OCCUPANCY DISTRIBUTION PREDICTION

Accounting for emergency risk, we model potential agent behaviors by extending occupancy prediction beyond agents' joint distribution to individual's short-term marginal distribution. As shown in the top of Fig. 2, the scene feature map $F_s$ integrate individual agent features $F_i \in F_A$ through an additional marginal occupancy encoder. $F_s$ attends only to the vectorized feature of a single agent, rather than all elements in $F_{vec}$, thereby capturing the agent's marginal behavioral feature $F_{m,i}$. Without modality decomposition, $F_{m,i}$ is directly leveraged to predict $i$-th agent's short-term marginal occupancy $\mathbf{O}_{m,i}$ of $T_s$ horizon ($T_s < T_f$) through upsampling and a 2D-CNN output head.

We perform agent pruning to reduce the computational cost, retaining only interactive agents for marginal occupancy distribution prediction. We devise a rule-based pruning method with minor inductive bias by selecting agents whose future bounding boxes cross with ego's future path. Please refer to Appendix. D for details about agent pruning. Marginal occupancy prediction is applied only in the training phase, enabling the model to learn individual agent's behavior patterns and short-term uncertainties during the optimization process, which also serves as a foundation for contingency planning. Its entire procedure is presented as follows:

$$\{F_i \,|i = 1, 2, ..., N_m\} = \mathrm{Prune}(F_A),$$
$$F_{m,i} = f_m(F_s, F_i), \tag{7}$$
$$\mathbf{O}_{m,i} = \mathrm{Conv}(\mathrm{Upsample}(F_{m,i})).$$

where $N_m$ indicates the number of pruned interactive agents, while $f_m$ denotes the network of marginal occupancy encoder, sharing the same structure as joint scene encoder.

## 3.4 TRAINING LOSS

Apart from basic trajectory and occupancy flow supervision, we devise a suite of specialized losses as shown in (a)-(c) in Fig. 3. Occupancy interference loss is applied to strengthen the interaction awareness between predicted ego and agents' joint occupancy. Occupancy guidance loss is further introduced to guide the trajectories with future occupancy probabilistic distribution. Utilizing predicted marginal and joint occupancy distribution, we apply contingency planning strategy for better driving safety without compromising efficiency. We leave details of trajectory planning loss $\mathcal{L}_{traj}$ and occupancy prediction loss $\mathcal{L}_{occ}$ in Appendix. E.

**Occupancy interference loss**. The occupied areas of ego vehicle and agents are mutually exclusive, inherently reflecting future interactions. We incorporate this property within occupancy interference loss, which is formulated as Eq. 8. For training stability, we apply teacher-forcing Williams & Zipser

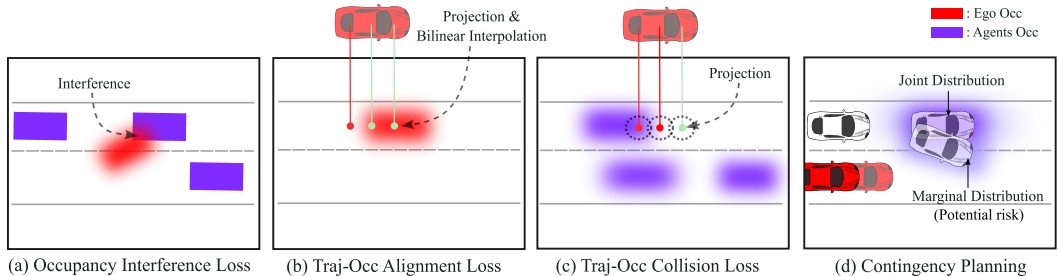

Figure 3: (a) - (c) illustrates the proposed losses: (a) penalizes the overlapping region between occupancy prediction and its opponent's ground truth; (b) is imposed if ego's trajectory point is mapped onto its low-occupancy region; (c) is applied when the distance between ego's mapped point and agents' high-occupancy region is smaller than the safety margin. (b) and (c) constitute occupancy guidance loss. (d) illustrates our contingency policy in occupancy space. Short-term occupancy takes the maximum value of joint and marginal distribution, considering potential behavior uncertainty.

(1989) technique. For both ego and agents, the loss is calculated as the average predicted occupancy within the opponents' GT occupied regions, measuring the extent of spatial interference. Optimizing occupancy interference loss effectively enhances the interaction awareness in the occupancy space, facilitating subsequent occupancy guidance loss as well.

$$\mathcal{L}_{oe} = \mathrm{sum}(\mathbf{O}_e^* \cdot \mathbf{O}_a^{gt}) \,/\, \mathrm{sum}(\mathbf{O}_a^{gt}), \;\; \mathcal{L}_{oa} = \mathrm{sum}(\mathbf{O}_a^* \cdot \mathbf{O}_e^{gt}) \,/\, \mathrm{sum}(\mathbf{O}_e^{gt}),$$
$$\mathcal{L}_{oi} = \mathcal{L}_{oe} + \mathcal{L}_{oa}, \tag{8}$$

where $\mathbf{O}^*$ denotes occupancy prediction of best modality and $\mathbf{O}^{gt}$ denotes ground truth occupancy.

**Occupancy guidance loss**. Future occupancy can serve as spatial prior information in the BEV view, explicitly guiding ego's trajectory planning. Utilizing both $\mathbf{O}_a^*$ and $\mathbf{O}_e^*$, we devise trajectory-occupancy collision loss to steer the ego trajectory away from agents' high-probability occupied area and trajectory-occupancy alignment loss is introduced to constrain ego trajectory within its own high-probability occupied area.

Considering ego vehicle's shape, at each timestep $t$, we offset trajectory position $(x_t, y_t)$ into $N_v$ circle centers $\{P_i^t \mid i = 1...N_v\}$ to approximate the ego vehicle. For trajectory-occupancy alignment loss, we obtain $P_i^t$'s occupied probability $O_i^t$ on the predicted ego occupancy grid through coordinate projection and bilinear interpolation. To enforce alignment between trajectory and high-occupancy areas, penalty is applied to points whose occupancy probability is below the predefined threshold $\varepsilon$.

$$\mathcal{L}_{align} = \frac{1}{T_f} \sum_{t=1}^{T_f} \sum_{i=1}^{N_v} \max(0, \varepsilon - O_i^t). \tag{9}$$

For trajectory-occupancy collision loss, after mapping $P_i^t$ onto the predicted agents' occupancy grid through coordinate projection, we compute its minimum Euclidean distance $d_i^t$ to high-occupancy regions (where occupancy probability exceeds threshold $\zeta$). Collision penalty is applied when $d_i^t$ is below the safety margin $\eta$, which indicates high collision risk.

$$\mathcal{L}_{collision} = \frac{1}{T_f} \sum_{t=1}^{T_f} \sum_{i=1}^{N_v} \max(0, \eta - d_i^t). \tag{10}$$

**Contingency Planning** is incorporated into $\mathcal{L}_{collision}$ to enhance planning safety. Conventional contingency planning divides trajectory into short-term safe maneuver and subsequent branched long-term behavior sets. Compared to trajectory, occupancy probability models behavioral uncertainty more effectively. In our work, the predicted short-term marginal occupancy $\mathbf{O}_{m,i}$ is leveraged to represent the uncertainty of single agent's short-term behavior. As shown in Eq. 11, before computing $\mathcal{L}_{collision}$, the predicted all-agents' occupancy $\mathbf{O}_a^*$ incorporates $\{\mathbf{O}_{m,i} \mid i = 1...N_m\}$ through an element-wise maximum operation over a short period $T_s$. Thus, $\mathcal{L}_{collision}$ enforces ego trajectory to account for short-term risks caused by agents' behavior uncertainty, while keeping modality-compliant planning in the long term.

$$\tilde{\mathbf{O}}_a^* = \begin{cases} \max(O_a^{t\,*}, \max\limits_{i=1}^{N_m}(O_{m,i}^t)), \; t \le T_s \\[2mm] O_a^{t\,*}, \; t > T_s \end{cases} \tag{11}$$

Table 1: Performance comparison of closed-loop planning on nuPlan **Val14** benchmark. All metrics are higher the better. Among learning-based methods, OccDriver achieves SOTA in both none-reactive score (NR-S) and reactive score (R-S) with top safety performance (Collisions and TTC).

| Type | Planner | Val14 | | | | | |
|------|---------|-----------|-----|---------|----------|------|------|
| | | Collisions | TTC | Comfort | Progress | NR-S | R-S |
| Expert | Log-Replay | 0.988 | 0.944 | 0.993 | 0.990 | 0.937 | 0.812 |
| Learning | PDM-Open Dauner et al. (2023) | 0.745 | 0.691 | 0.995 | 0.699 | 0.502 | 0.548 |
| | RasterModel Caesar et al. (2021) | 0.870 | 0.815 | 0.815 | 0.806 | 0.669 | 0.647 |
| | UrbanDriver Scheel et al. (2022) | 0.856 | 0.803 | **1.000** | 0.808 | 0.677 | 0.649 |
| | PlanTF Cheng et al. (2024b) | 0.941 | 0.907 | 0.937 | **0.898** | 0.853 | 0.771 |
| | PLUTO Cheng et al. (2024a) | 0.962 | 0.933 | 0.964 | 0.896 | 0.890 | 0.800 |
| | BeTopNet Liu et al. (2024a) | 0.966 | 0.916 | 0.932 | 0.866 | 0.883 | 0.837 |
| | DiffusionPlanner Zheng et al. (2025a) | - | - | - | - | **0.899** | 0.828 |
| | OccDriver (Ours) | **0.971** | **0.938** | 0.969 | 0.885 | 0.896 | **0.838** |

Table 2: Performance comparison of closed-loop planning on nuPlan **Test14 − Hard** benchmark. All metrics are higher the better. Compared to vectorized-only, topology-guided and diffusion-based methods, OccDriver achieves top driving scores with desirable planning safety and progress.

| Planner | Inference Time (ms) | Test14-Hard | | | |
|---------|---------------------|-------------|----------|------|-----|
| | | Collisions | Progress | NR-S | R-S |
| PLUTO Cheng et al. (2024a) | 7.39 | 0.938 | 0.816 | 0.787 | 0.753 |
| BeTopNet Liu et al. (2024a) | 70.00 | **0.968** | 0.747 | 0.771 | 0.688 |
| DiffusionPlanner Zheng et al. (2025a) | 40.00 | - | - | 0.760 | 0.692 |
| OccDriver (Ours) | 23.03 | 0.941 | **0.829** | **0.794** | **0.759** |

Occupancy guidance loss $\mathcal{L}_{og}$ is represented as a weighted sum of $\mathcal{L}_{align}$ and $\mathcal{L}_{inter}$, regulating ego's trajectory with the explicit guidance of predicted future scenario:

$$\mathcal{L}_{og} = w_1 \mathcal{L}_{align} + w_2 \mathcal{L}_{collision}. \tag{12}$$

All losses are differentiable, allowing for end-to-end training. The overall training loss comprises trajectory planning loss $\mathcal{L}_{traj}$, occupancy prediction loss $\mathcal{L}_{occ}$, occupancy interference loss $\mathcal{L}_{oi}$ and occupancy guidance loss $\mathcal{L}_{og}$. It is formulated as :

$$\mathcal{L} = \mathcal{L}_{traj} + \mathcal{L}_{occ} + \mathcal{L}_{oi} + \mathcal{L}_{og}. \tag{13}$$

## 4 EXPERIMENTS

### 4.1 BENCHMARK AND METRICS

OccDriver is trained and evaluated on nuPlan dataset Caesar et al. (2021). We use a standardized training set of 1M frames with 2s history and 8s horizon. Our evaluation is conducted on **Val14** split Dauner et al. (2023), and verified using closed-loop evaluation metrics: Non-Reactive Closed-Loop Simulation(CLS-NR) score and Reactive Closed-Loop(CLS-R) provided by nuPlan simulator. To verify performance under challenging cases, we further evaluate OccDriver on **Test14 − Hard** split Cheng et al. (2024c). We follow the nuPlan challenge framework and report the official Planning Scores. More experiment details and parameter settings are in Appendix. F.

### 4.2 MAIN RESULTS

**Comparison with State of the Art.** We conduct a comparative analysis between OccDriver and existing state-of-the-art learning-based methods on nuPlan **Val14** benchmark. All methods are evaluated without post-processing to compare models' planning performance. Comparative results are presented in Table. 1. In closed-loop simulation, OccDriver gains SOTA planning scores of 0.896

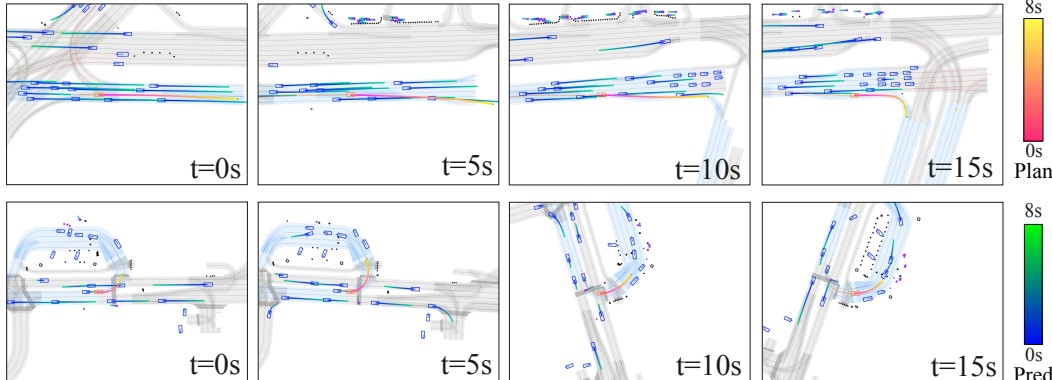

Figure 4: Qualitative results of closed-loop planning. Each scenario lasts 15 seconds. OccDriver performs interaction-compliant planning in changing lane in dense traffic (first row) and turning left after yielding (second row).

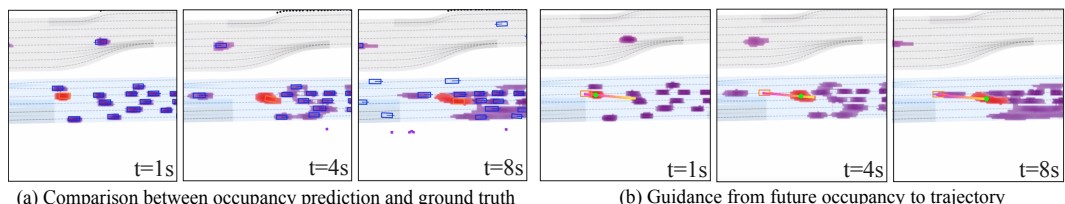

(a) Comparison between occupancy prediction and ground truth    (b) Guidance from future occupancy to trajectory

Figure 5: Visualization of future occupancy prediction and guidance. (a) ego's (red) and agents' (purple) occupancy predictions coincide with their GT bounding boxes; (b) planning trajectory (green point) aligns with ego's occupancy while keeps away from agents' occupancy.

NR-Score and 0.838 R-Score. Notably, it demonstrates best performance on safety metircs, boosting to 0.971 (Collisions) and 0.938 (TTC). This can be attributed to guidance of future occupancy and contingency planning strategy, improving ego's awareness of spatial interactions and behavioral uncertainty. Unlike previous methods, OccDriver enhances safety performance with less degradation on Comfort and Progress. This results from the iterative decoding process, which refines trajectory holistically and generates scenario-compliant behavior.

To further demonstrate the advantages of OccDriver under challenging scenarios, we conduct comparison with vectorized-only method (PLUTO), topology-guided method (BeTopNet) and diffusion-based method (Diffusion Planner) on $\mathbf{Test14 - Hard}$ benchmark, as presented in Table. 2. OccDriver reaches the highest driving scores in both non-reactive and reactive simulation. Compared to PLUTO, OccDriver achieves enhancement in both safety and progress metics, demonstrating the effectiveness of incorporating occupancy branch. OccDriver also outperforms BeTopNet with +3.0% NR-S and +10.3% R-S, which models multi-agent interactions with topology connections. Despite excelling in safety metrics, BeTopNet suffers from a substantial degradation in progress. In contrast, our method maintains a favorable trade-off between safety and progress, leading to the best driving scores. This suggests that explicit spatial occupancy delivers more intuitive and fine-grained interaction relations than implicit topological structures, avoiding over-conservative planning. Results in Fig. 4 further prove our model's robust performance in tackling dense traffic scenarios and multi-agent interactive scenarios. Notably, compared to denoising paradigm of diffusion and the intricate topological modeling, OccDriver has less inference latency, enabling deployment in real-world applications.

**Qualitative results.** To visually demonstrate the effectiveness of future occupancy guidance, we render both the predicted future occupancy and ego's planning trajectory simultaneously. As depicted in Fig. 5, predicted occupancy overlaps with ground-truth bounding boxes, demonstrating the accuracy of future occupancy prediction and robustness of scene evolution reasoning. Besides, planning trajectory is positioned within the ego's high-occupancy region and steers away from agents' high-occupancy region. This further validates effective guidance from future occupancy via implicit feature aggregation and explicit losses supervision.

Table 3: Ablation results of OccDriver's planning performance with different components. All proposed components contribute to improvements in safety metrics and driving scores. $\mathcal{L}_{oi}$, $\mathcal{L}_{collision}$, $\mathcal{L}_{align}$ denotes occ interference loss, traj-occ collision loss and traj-occ alignment loss. $\mathcal{MP}$ and $\mathcal{CP}$ denotes marginal prediction and contingency planning.

| $\mathcal{MP}$ | $\mathcal{L}_{oi}$ | $\mathcal{L}_{collision}$ | $\mathcal{L}_{align}$ | $\mathcal{CP}$ | Collisions | TTC | Comfort | Progress | NR-Score | R-Score |
|---|---|---|---|---|---|---|---|---|---|---|
| - | - | - | - | - | 0.933 | 0.905 | 0.966 | **0.893** | 0.859 | 0.787 |
| ✓ | - | - | - | - | 0.938 | 0.913 | 0.968 | 0.889 | 0.863 | 0.800 |
| ✓ | ✓ | - | - | - | 0.943 | 0.914 | **0.975** | 0.886 | 0.864 | 0.807 |
| ✓ | ✓ | ✓ | - | - | 0.960 | 0.931 | 0.966 | 0.882 | 0.879 | 0.825 |
| ✓ | ✓ | ✓ | ✓ | - | 0.960 | 0.928 | 0.971 | 0.892 | 0.885 | 0.830 |
| ✓ | ✓ | ✓ | ✓ | ✓ | **0.971** | **0.938** | 0.969 | 0.885 | **0.896** | **0.838** |

Table 4: Impact of different horizon $T$ of joint occupancy in occupancy guidance loss.

| $T$ | Collisions | TTC | NR-score | R-score |
|---|---|---|---|---|
| 2s | 0.933 | 0.908 | 0.828 | 0.787 |
| 4s | 0.943 | 0.916 | 0.843 | 0.799 |
| 6s | **0.946** | 0.914 | **0.845** | **0.801** |
| 8s | 0.931 | **0.917** | 0.828 | 0.787 |

Table 5: Impact of different threshold $\zeta$ of high-occupancy regions in $\mathcal{L}_{collision}$

| $\zeta$ | Collisions | TTC | NR-score | R-score |
|---|---|---|---|---|
| 0.5 | 0.938 | 0.916 | 0.837 | 0.786 |
| 0.6 | 0.941 | **0.917** | 0.841 | 0.799 |
| 0.7 | 0.946 | 0.914 | **0.845** | **0.801** |
| 0.8 | **0.948** | 0.903 | 0.840 | 0.799 |

## 4.3 ABLATION STUDY

To investigate the effectiveness of proposed framework, auxiliary losses and contingency planning strategy in our work, we conduct an ablation study. All ablation experiments are evaluated on **Val14** benchmark, and the results is shown in Table. 3. We first evaluate base dual-branch framework without marginal occupancy prediction module, achieving near-SOTA performance with 0.859 NR-S and 0.787 R-S. With the integration of marginal occupancy prediction module, both safety metrics and driving scores exhibit improvements. It is attributed to the modeling of individual behavior patterns and short-term uncertainty.

Based on proposed framework, we first apply occupancy interference loss $\mathcal{L}_{oi}$ to learn exclusivity between ego's and other agents' occupancy. It enhances Collision metric while benefiting Comfort metric due to improved awareness of spatial interaction and feasible safe areas. In the fourth experiment, a substantial boost in safety metrics (Collision from 0.943 to 0.960 and TTC from 0.914 to 0.931) is observed after adding $\mathcal{L}_{collision}$ into training, attributed to its explicit penalty for collision and excessive proximity to other agents. On this basis, we further introduce $\mathcal{L}_{align}$, which yields an improvement in Progress metric due to ego occupancy's positive guidance, jointly optimizing safety and efficiency. Notably, the continuous increases of driving safety do not compromise the Comfort metric. The above experiments demonstrate that the proposed losses effectively distill spatial information from occupancy branch to trajectory branch.

The bottom row of Table. 3 presents the ablation study on contingency planning. Safety metrics and driving scores reach their peaks at a minor trade-off in the Progress metric. We attribute this to the model's consideration of potential marginal behaviors of relevant agents, adopting a relatively cautious strategy to boost contingency safety.

Table. 4 shows the effects of different horizons of joint occupancy prediction in occupancy guidance loss. We observe a consistent improvement in Collision metric and driving scores as the horizon $T$ increases, reaching peak at $T = 6$s. Driving performance starts to degrade when the horizon extends to 8s. It suggests that long-horizon future occupancy captures scene dynamics and agents' long-term interactions, facilitating interaction-consistent planning. However, as the uncertainty of occupancy prediction accumulates over time, leveraging highly uncertain future occupancy as guidance introduces ambiguous or incorrect supervision signal, ultimately degrading planning performance.

Table 6: Planning performance comparison between coarse and fine trajectories.

| Trajectory | Collision | TTC | NR-S | R-S |
|---|---|---|---|---|
| Coarse | 0.931 | 0.903 | 0.836 | 0.790 |
| Fine | **0.946** | **0.914** | **0.845** | **0.801** |

Table 7: Impact of agent pruning on closed-loop planning. $\mathcal{AP}$ denotes agent pruning.

| $\mathcal{AP}$ | Collision | TTC | Progress | NR-S |
|---|---|---|---|---|
| without | **0.951** | **0.915** | 0.866 | 0.840 |
| with | 0.946 | 0.914 | **0.873** | **0.845** |

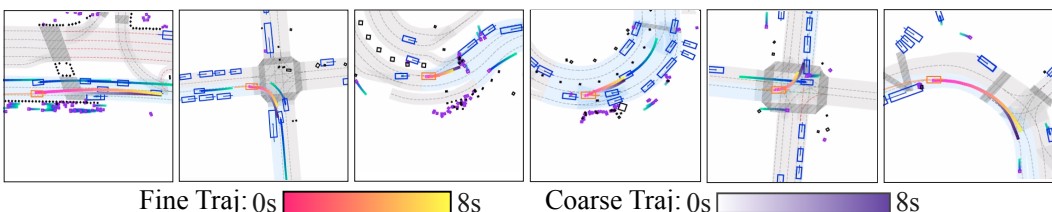

Fine Traj: 0s ▨ 8s    Coarse Traj: 0s ▨ 8s

Figure 6: Visualization of coarse trajectory and fine trajectory in nuPlan **Test14 − Hard** benchmark.

In Table. 5, we ablate different thresholds of high-occupancy regions in $\mathcal{L}_{collision}$. Collision metric increases as the threshold $\zeta$ rises, while TTC starts decreasing once $\zeta$ exceeds 0.6. The reason is that higher threshold offers more deterministic collision supervision, while simultaneously losing uncertainty awareness. Overall, $\zeta = 0.7$ reaches the optimal performance.

Table. 6 compares the planning metrics between coarse and fine trajectories. The quantitative result shows that fine trajectories achieve improved safety metrics and driving scores over coarse trajectories. It demonstrates that the integration of future scene evolution, together with explicit occupancy guidance, provides positive information for trajectory planning, enabling refinement into safer trajectories. Fig. 6 exhibits qualitative comparison between coarse and fine trajectories on **Test14 − Hard** set. In interactive scenarios, compared with collision-prone coarse trajectories, fine trajectories are evidently altered to avoid collisions and handle interactions effectively, validating the effectiveness of the coarse-to-fine decoding architecture and occupancy guidance.

The ablation results of agent pruning is shown in Table. 7. Because of GPU memory constraint, we select 8 nearest non-interactive agents, together with interactive agents (kept in agent pruning), for marginal occupancy prediction. The results indicate a minor drop in safety performance after agent pruning, validating the effectiveness of proposed pruning in keeping interaction-relevant agents. With agent pruning, our model attains a favorable balance between safety and progress. More importantly, agent pruning substantially reduces the computational overhead of marginal prediction, thereby improving the model's scalability. In Appendix. H, we analyze the impact of agent count on memory usage. The near-linear scaling property justifies the necessity of filtering irrelevant agents via pruning.

## 5 CONCLUSION

In this paper, we present OccDriver, a future occupancy guided dual-branch trajectory planning framework. Occupancy branch is incorporated to predict future scenes in occupancy space, guiding interaction-aware trajectory planning through implicit iterative decoding process and explicit loss supervision. Contingency planning is applied, leveraging short-term marginal and long-term joint occupancy predictions simultaneously to mitigate uncertainty risks and sustain scene consistency. Experiments on nuPlan benchmark verify OccDriver's state-of-the-art performance in closed-loop planning, leading to significant improvements in driving safety.

ACKNOWLEDGMENTS

We thank the anonymous reviewers for their constructive suggestions. This work is supported by the National Natural Science Foundation of China (No. 62476192) and the WDZC program of BIAM (Grant No. 2019-363), in collaboration with SenseAuto. Any opinions, findings, conclusions or recommendations expressed in this paper are those of the authors and do not reflect the views of the National Natural Science Foundation of China or BIAM.

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

APPENDIX

## A  DISCUSSIONS

Towards a better understanding of this work, we supplement intuitive questions that may arise and provide empirical answers.

### A.1  WHAT IS THE KEY OF OCCDRIVER AS A WORLD MODEL GUIDED PLANNING PARADIGM?

Most world models in autonomous driving predict future scene evolution by generating videosHu et al. (2023a); Wang et al. (2024a); Yang et al. (2024); Zhao et al. (2025) or 3D representationsZhang et al. (2023); Zheng et al. (2024); Li et al. (2025), serving primarily as data generatorsMa et al. (2025) and RL simulation environmentsLi et al. (2024a); Gao et al. (2025). DriveWorldMin et al. (2024) and LAWLi et al. (2024b) leverage world models to extract latent scene representations, enhancing spatiotemporal scene understanding of autonomous driving systems. Recent research integrates world models into end-to-end planning. MILEHu et al. (2022a) and DriveDreamerWang et al. (2024a) jointly decode driving actions and the corresponding future scene dynamics. Drive-WMWang et al. (2024b)and Drive-OccWorldYang et al. (2025) utilize explicit future scene predictions to formulate cost functions for trajectory selection or optimization. Latent feature-based methods implicitly transfer future world knowledge to the planner. LatentDriverXiao et al. (2025) unifies the environment's next states and ego's next action as a mixture distribution. World4DriveZheng et al. (2025b) constructs an intention-aware latent world model to rank trajectories.

Compared with prior work, OccDriver extends beyond predicting future scenario evolution by explicitly exploiting interaction modeling in occupancy world and how this can benefit planning safety. OccDriver extracts spatial exclusivity and behavior uncertainty from ego and agents' occupancy distribution, distilling this interaction knowledge to trajectory planning via both feature transfer and occupancy guidance losses.

### A.2  WHAT ARE THE KEY DIFFERENCES BETWEEN OCCDRIVER AND OTHER OCCUPANCY-ASSISTED PLANNING METHODS?

Yang et al. (2025) proposed a 3D occupancy-based world model to evaluate the trajectories, which decoupled from the planner as a separate model. In contrast, the occupancy branch in OccDriver is integrated as part of the planner, where the planning branch incorporates occupancy feature during both training and inference, and the predicted occupancy is leveraged to design training losses that improve the planner's trajectory quality. Compared to Liu et al. (2023b), which only incorporates historical occupancy features to provide supplementary perception information, our approach further exploits future occupancy predictions to explicitly model future interactions, and integrate this interaction knowledge into trajectory decoding through both feature-level fusion and specialized loss supervision.

In our work, occupancy assistance yields benefits in two key aspects: information complementation and interaction modeling. On one hand, the incorporation of scene-level occupancy information supplements individual features with global scene context and spatial dynamics. On the other hand, we employ occupancy as explicit interaction modeling to guide safer planning. The probabilistic nature of occupancy represents behavioral uncertainty more effectively than ground-truth or trajectory regression. Marginal prediction and contingency planning further enhance the planner's risk awareness.

### A.3  WHY IS CONTINGENCY PLANNING SUITABLE FOR LEARNING-BASED INTERACTION MODELING, MAKING OCCUPANCY-GUIDED PLANNING SUPERIOR IN SAFETY?

Previous works enhance multi-agent prediction using game theoretic approaches and integrated prediction and planningLiu et al. (2024b) and conditional prediction. Although capable of generating scene compliant scene evolution, these methods suffer from over-optimistic when uncooperative agent behaviour occurs. Contingency planning is one of the strongest optimization-based tools for safety-critical motion planning. Yet, it suffers from discretized artifacts and the difficulty of enjoying

the power of data-driven planning. OccDriver proposes a natural paradigm to integrate contingency objective with learning-based algorithm, and mitigate the discretized artifacts by representing contingency objective within occupancy space.

### A.4 WHAT WOULD BE THE BROADER IMPACT OF OUR PARADIGM?

The paradigm that combines occupancy-based world model and contingency planning can be extended to a varies of sota methods. For example, we use regression method for trajectory learning to better compare with several strong baselines, but diffusion-based planner has shown superior performance by avoiding learning local average behavior. Our pipeline can be adapted to diffusion-based planner by introducing occupancy as classifier guidance and including occupancy feature as DiT condition. Previous works also exploit RL fine-tuning by collecting rollouts and rewards within a world model. Occupancy-based world models are often used as reward models and should be adaptable to RL paradigm. However, the actual gain and design of such architectures need further investigation.

## B   LIMITATION AND FUTURE WORK

Our current approach discretizes the occupancy distribution along the time dimension, lacking supervision across time intervals. Future work will extend this to a unified spatiotemporal occupancy distribution, ensuring smoother temporal consistency. Additionally, we aim to develop an end-to-end framework that seamlessly integrates information between branches. Furthermore, we will explore safe reinforcement learning for contingency planning, leveraging risk-aware optimization objectives such as Conditioned Value at Risk (CVaR).

## C   ARCHITECTURAL DETAILS

**Vectorized Encoder.** Ego's and agents' inputs involves kinematic states for past $T_h = 2$ seconds at 10 Hz and attributes. We only leverage the current state for ego vehicle to prevent shortcut bias. All scene inputs $\{\mathbf{E}, \mathbf{A}, \mathbf{S}, \mathbf{M}\}$ are firstly embedded to hidden dimension $D = 128$ with separate encoders. Ego state $\mathbf{E}$ is embedded through attention based State Dropout Encoder (SDE). Agents' states $\mathbf{A}$ are embedded through attention-based Feature Pyramid Network (FPN). Static objects $\mathbf{S}$ and map polylines $\mathbf{M}$ are embedded through 2-layer Multi Layer Perceptron (MLP) and PointNet-based vector encoder respectively. All embedded features are concatenated together and added respective positional embedding, as $F_{vec} \in \mathbb{R}^{(1+N_A+N_S+N_M) \times D}$. As depicted in Fig, a 4-layer transformer encoder is employed for feature fusion between scene elments. Each layer consists of a multi-head self-attention and a feedforward neural network (FFN).

**Rasterized Encoder.** We build a $H \times W = 128 \times 128$ spatial grid, covering the ego vehicle's current position with a range of [-20m, 60m] in the x-direction and [-40m, 40m] in the y-direction (resolution rate = 0.625m / pixel). Follow the practice of Ettinger et al. (2021), we seperately rasterize ego state, agent states and map into occupancy grids $\{\mathbf{O}_e, \mathbf{O}_a, \mathbf{O}_m\}$. The backward flow $\mathbf{FL}$ is constructed by calculating the displacement of the occupancy pixels between two consecutive timesteps for the same agent. We only use current occupancy grids and backward flow as input which are embedded and down-sampled to $F_o, F_f \in \mathbb{R}^{(32 \times 32 \times 128)}$ with 2 separate Swin-transformer encodersLiu et al. (2021b). Each swin-transformer encoder is a 2-layer transformer with window self-attention (W-SA) and shifted window self-attention (SW-SA). $F_o$ and $F_f$ are concatenated and fed into a MLP to form spatial scene feature $F_s \in \mathbb{R}^{(32 \times 32 \times 128)}$.

**Joint Scene Encoder** is a two-layer transformer module, which takes scene feature map $F_s$ as input query, deriving encoded occupancy feature $F_{occ}$. Each layer consists of a self-attention, a cross-attention ($F_{vec}$ as Key and Value), and a feed-forward network (FFN). **Marginal Occupancy Encoder** shares the same structure with Joint scene Encoder. The only difference is that the cross-attention in marginal occupancy encoder queries individual agent feature $F_i \in F_A$ instead of $F_{vec}$.

**Iterative Decoding Structure** is composed of 3 iterative decoders: coarse trajectory decoder $D_c$, future scene decoder $D_s$ and fine trajectory decoder $D_f$. In trajectory branch, we initialize $M = 6$ learnable embeddings $Q_m \in \mathbb{R}^{M \times D}$ to model ego's $M$ longitudinal modalities. $Q_m$ is concatenated with $F_E \subset F_{vec}$ and projected as $Q'_m \in \mathbb{R}^{M \times 1 \times D}$ via a MLP. For joint prediction, $Q'_m$ is further

concatenated with $F_A \subset F_{vec}$, forming $Q_{vec} \in \mathbb{R}^{M \times (N_A+1) \times D}$. In occupancy branch, $D_s$ directly utilizes $F_{occ}$ as decoding query. $D_c$ and $D_f$ share the same 4-layer Transformer decoder structure, with each layer consisting of a self-attention, two cross-attentions, and a feed-forward network (FFN). We implement $D_s$ as a 2-layer Transformer decoder. Each layer of $D_s$ contains a self-attention, a cross-attention and an FFN.

**Prediction Heads.** Deocoded fine trajectory query $Q_f$ is passed through 2 separate MLPs to generate joint prediction trajectories $\mathbf{Y} \in \mathbb{R}^{M \times (N_A+1) \times (T_f/\Delta_{traj}) \times 6}$ for future $T_f = 8$ seconds and corresponding confidence scores $\pi \in \mathbb{R}^{M \times 1}$. Trajectory states contains $(x, y, cos(\theta), sin(\theta), v_x, v_y)$. Coarse trajectory query $Q_c$ is processed with another MLP to generate coarse trajectories $\mathbf{Y}_c$. Future scene feature $Q_s$ is up-sampled and passes through two separate 2d-CNNs to output future occupancy grids $\mathbf{O} = \{\mathbf{O}_e, \mathbf{O}_a\} \in \mathbb{R}^{M \times (T_f/\Delta_{occ}) \times 128 \times 128 \times 2}$ and flow grids $\mathbf{FL} \in \mathbb{R}^{M \times (T_f/\Delta_{occ}) \times 128 \times 128 \times 2}$. Agents' marginal occupancy features $\{f_{m,i}\}$ is decoded into short-term marginal occupancy $\{\mathbf{O}_{m,i} \mid i = 1, 2, ..., N_m\} \in \mathbb{R}^{N_m \times (T_s/\Delta_{occ}) \times 128 \times 128 \times 1}$ for $T_s = 2$s. $\Delta_{traj}$ and $\Delta_{occ}$ indicate the prediction frame intervals of trajectory and occupancy, respectively.

## D AGENT PRUNING

Although OccDriver reduces the computational complexity of contingency planning from $O(n^2)$ to $O(n)$, calculating marginal occupancy distribution of each agent is unacceptable. Therefore, we propose a rule-based agent pruning method that identifies non-directly-interactive agents according to their ground truth trajectories. During training, the annotated non-interactive agents are directly ignored in marginal occupancy computation, and an agent role prediction head takes in agent features and predicts the confidence score of the role of each agent. During inference, the agents that have above threshold confidence of being non-interactive are pruned in marginal occupancy computation.

We identify 5 regular agent roles: in-lane leader, in-lane follower, lateral intruder(cut-in, cut-out, crossing agents), overtaking target, and non-interactive agents. The interactive relationship is determined by examining whether the ground truth bboxes of ego and agent intersect within a time difference of 4s. If ego reaches the intersection position prior to the agent, the agent is either an in-lane follower or an overtaking target, which is then determined by their heading difference at the intersection position. Similarly, if ego reaches the intersection position later than the agent, the agent is either an in-lane leader or a lateral intruder, which is also further determined by their heading difference at the intersection position. All other agents are considered non-interactive agents as they have no direct impact on the ego vehicle in the short term, thus can be ignored for marginal occupancy computation. Note that predicting detailed agent roles instead of only interactive and non-interactive agents alleviates the causal confusion problem, e.g., in-lane follower.

## E TRAINING LOSS

**Trajectory planning loss**. To avoid mode collapse Liu et al. (2021a), we employ teacher-forcing Williams & Zipser (1989) technique during the training process. We compute the length of the ground-truth trajectory and assign it to the corresponding longitudinal modality based on 20 m segments. The trajectory $y^*$ of the target modality is used to compute the regression loss. For regression loss $\mathcal{L}_{traj}$, we employ the smooth L1 loss, and for classification loss $\mathcal{L}_{cls}$, we utilize the cross-entropy loss.

To facilitate future scene prediction with more reliable initial behaviors and accelerate training convergence, we apply smooth L1 loss $\mathcal{L}_{coarse}$ to coarse trajectory $y_c^*$. The overall trajectory planning loss $\mathcal{L}_{traj}$ is formulated as a weighted sum of its components:

$$\mathcal{L}_{traj} = w_3 \mathcal{L}_{reg} + w_4 \mathcal{L}_{cls} + w_5 \mathcal{L}_{coarse}. \tag{14}$$

**Occupancy prediction loss**. We supervise both joint occupancy prediction of target modality $\mathbf{O}_e^*, \mathbf{O}_a^*$ and marginal occupancy predictions $\{\mathbf{O}_{m,i} \mid i = 1, 2, ..., N_v\}$. Due to the significantly larger proportion of unoccupied regions, we utilize focal loss Ross & Dollár (2017) as loss function $\mathcal{L}_e$, $\mathcal{L}_a$, $\mathcal{L}_m$ . Additionally, L1 loss $\mathcal{L}_{flow}$ is applied to supervise flow prediction $\mathbf{FL}^*$, facilitating better understanding of traffic scene's dynamics. The overall occupancy prediction loss $\mathcal{L}_{occ}$ is defined as:

$$\mathcal{L}_{occ} = w_6 \mathcal{L}_e + w_7 \mathcal{L}_a + w_8 \mathcal{L}_m + w_9 \mathcal{L}_{flow}. \tag{15}$$

Table 8: Loss weight parameters.

| Loss | $\mathcal{L}_{reg}$ | $\mathcal{L}_{cls}$ | $\mathcal{L}_{coarse}$ | $\mathcal{L}_e$ | $\mathcal{L}_a$ | $\mathcal{L}_m$ | $\mathcal{L}_{flow}$ | $\mathcal{L}_{align}$ | $\mathcal{L}_{collision}$ |
|---|---|---|---|---|---|---|---|---|---|
| **Weight** | 4 | 1 | 0.3 | 3000 | 800 | 2000 | 1 | 1 | 9 |

Table 9: Hyperparameters and configuration settings.

| Notation | Parameters | Values |
|---|---|---|
| $M$ | Number of modality | 6 |
| $T_f$ | Horizon of joint prediction | 8s |
| $T_s$ | Horizon of marginal prediction | 2s |
| $\Delta_{traj}$ | Frame interval of trajectory | 0.1s |
| $\Delta_{occ}$ | Frame interval of occupancy | 1s |
| $N_v$ | Number of covering circles | 3 |
| $\eta$ | Safety margin in $\mathcal{L}_{collision}$ | 2m |
| $\varepsilon$ | Occupancy threshold in $\mathcal{L}_{align}$ | 0.8 |
| $\zeta$ | Occupancy threshold in $\mathcal{L}_{collision}$ | 0.7 |

Together with $w_1$ and $w_2$ for $\mathcal{L}_{align}$ and $\mathcal{L}_{collision}$, detailed loss weight parameters are shown in Table. 8.

# F EXPERIMENTAL DETAILS

We perform training on 32 NVIDIA RTX 4090 GPUs with a batch size of 16 for 30 epochs. It takes 20 hours for finishing training. The total number of parameters is 7.9 M and the model size is 31 MB. Learning rate is set to 1e-3, with 3 warm-up epochs and Cosine scheduler. We adopt AdamW optimizer, applying a weight decay of 0.0001. For proper guidance and training stability, we add occupancy guidance loss after 15 epochs. In occupancy guidance loss, the gradient of occupancy predictions is detached to guarantee unidirectional guidance. Details on parameter settings can be found in Table. 9.

Table 10: Impact of different prediction horizons $T_{occ}$ of joint occupancy.

| $T_{occ}$ | Collisions | TTC | NR-score | R-score |
|---|---|---|---|---|
| 2s | 0.925 | 0.886 | 0.836 | 0.774 |
| 4s | 0.928 | 0.886 | 0.839 | 0.778 |
| 6s | 0.933 | **0.901** | **0.848** | 0.779 |
| 8s | **0.935** | 0.900 | 0.846 | **0.782** |

Table 11: Impact of different weights $w_2$ of trajectory-occupancy collision loss $\mathcal{L}_{collision}$

| $w_2$ | Collisions | TTC | Progress | NR-score |
|---|---|---|---|---|
| 3 | 0.932 | 0.903 | 0.876 | 0.840 |
| 6 | 0.943 | 0.908 | 0.869 | 0.843 |
| 9 | **0.946** | **0.914** | 0.873 | **0.845** |
| 12 | 0.922 | 0.879 | **0.902** | 0.826 |

Table 12: Impact of different numbers $M$ of decoding modalities.

| $M$ | Collisions | TTC | Comfort | Progress | NR-S | R-S |
|---|---|---|---|---|---|---|
| 1 | 0.915 | 0.887 | 0.975 | 0.860 | 0.823 | 0.755 |
| 3 | 0.931 | 0.904 | **0.978** | 0.869 | 0.835 | 0.784 |
| 6 | 0.946 | **0.914** | 0.976 | 0.873 | 0.845 | 0.801 |
| 9 | **0.951** | 0.911 | 0.972 | 0.876 | **0.848** | **0.805** |
| 12 | 0.944 | 0.898 | 0.965 | **0.882** | 0.840 | 0.797 |

Table 13: Comparison between vector-only and dual-branch architectures.

| Framework | Collision | TTC | Comfort | Progress | NR-S | R-S |
|---|---|---|---|---|---|---|
| Vec branch | 0.917 | 0.884 | **0.976** | 0.885 | 0.836 | 0.769 |
| Dual branch | **0.933** | **0.905** | 0.966 | **0.893** | **0.859** | **0.787** |

## G    ADDITIONAL ABLATION STUDIES

For efficiency, we sample $20\%$ of **nuPlan** training set for hyperparameter ablation experiments and conduct evaluation on the **Val14** split.

**Impact of dual-branch architecture.** Table. 13 presents the ablation results for dual-branch architecture. The dual-branch architecture achieves superior safety and progress metrics compared to the vectorized-only baseline, demonstrating the effectiveness of proposed dual-branch structure. We attribute the enhanced planning performance to incorporating both current and future occupancy features from the occupancy branch. Current occupancy feature supplements global spatial information lost in the vectorized representation. Future occupancy feature further extracts scene-evolution information to refine scene-consistent trajectories.

**Different prediction horizons of joint occupancy.** To further investigate the influence of occupancy prediction on trajectory planning, we ablate different joint occupancy prediction horizons $T_{occ}$ without incorporating auxiliary losses. As shown in Table. 10, we observe that increasing $T_{occ}$ leads to simultaneous improvements in both safety metrics and closed-loop driving scores. It suggests that extended occupancy predictions captures long-term dynamics of the traffic, enabling more consistent spatial guidance for trajectory planning. Further performance improvement ceases when $T_{occ}$ extends to 6s, presumably due to the compounding uncertainty inherent in long-term occupancy predictions. In practice, $T_{occ}$ is set as $T_f = 8s$ to ensure consistency with trajectory planning.

**Different weights of trajectory-occupancy collision loss.** Table. 11 presents the effects of varying $w_2$ aligned to $\mathcal{L}_{collision}$ on driving performance. As $w_2$ increases from 3 to 9, OccDriver consistently improves driving safety ($+1.4\%$ Collision and $+1.2\%$ TTC) with minor degradation in Progress, validating the effectiveness of $\mathcal{L}_{collision}$ in collision avoidance. However, further increasing $w_2$ to 12 results in significant performance degradation, due to over-penalization of collisions, which leads to undesirable shortcut that ego moves out of the occupancy field quickly to avoid collision supervision.

**Different numbers of modalities.** As presented in Table. 12, M=9 yields optimal driving scores(0.848 NR-S and 0.805 R-S). It indicates that M=9 is sufficient to cover the ego's longitudinal behaviors. More modality enables finer-grained division of ego's behavioral patterns, leading to an continuous growth of Progress metric. Both Collision and TTC degrade when M exceeds 9, which we attribute to overly redundant modality partitioning that reduces trajectory flexibility. Besides, the increased number of modalities reduces the training data allocated to each modality, eventually causing insufficient training and degraded trajectory quality.

## H    COMPUTATIONAL COST ANALYSIS

Table 14: Impact of different components on the training computational overhead.

| $\mathcal{Base}$ | $\mathcal{OP}$ | $\mathcal{MP}$ | Latency (ms) | Params (M) |
|---|---|---|---|---|
| ✓ | - | - | 29.77 | 3.8 |
| ✓ | ✓ | - | 60.61 | 7.4 |
| ✓ | ✓ | ✓ | 69.24 | 7.9 |

Table 15: Scalability of memory usage with respect to agent count in marginal prediction.

| Agent Num (bs=8) | GPU Memory (GB) |
|---|---|
| 0 | 3.67 |
| 5 | 5.36 |
| 10 | 7.98 |
| 20 | 13.22 |

The training cost breakdown of our model is reported in Table. 14. $\mathcal{OP}$ and $\mathcal{MP}$ denote occupancy prediction and marginal prediction, respectively. According to the table, the total computational costs of our model is relatively low. Occupancy decoding and guidance account for nearly half of

Table 16: Performance comparison of end-to-end planning on Navsim test split. All metrics are higher the better. OccDriver achieves safe and efficient planning under end-to-end paradigm.

| Method | NC | DAC | TTC | CF | EP | PDM-S |
|---|---|---|---|---|---|---|
| UniAD | 97.8 | 91.9 | 92.9 | 100 | 78.8 | 83.4 |
| Transfuser | 97.9 | 92.8 | 92.8 | 100 | 79.2 | 84.0 |
| PARA-Drive | 97.9 | 92.4 | 93.0 | 99.8 | 79.3 | 84.0 |
| Baseline | 98.1 | 95.7 | **94.8** | 100 | 82.6 | 87.7 |
| OccDriver | **98.4** | **97.5** | 94.3 | **100** | **84.0** | **89.2** |

the total parameters and training latency, due to the rasterized feature representation and additional losses. Marginal distribution prediction introduces minimal computational overhead, a result of its lightweight model architecture.

In Table. 15, we analyze scalability of the number of agents used in marginal prediction. A consistent increase in GPU memory allocation was observed as the agent count grows. Predicting marginal occupancy distributions for all agents incurs computationally prohibitive memory overhead. To address this issue, We leverage agent pruning to keep only interactive agents (typically fewer than 10), significantly enhancing scalability.

## I  TRANSITIONING TO END-TO-END PARADIGM

We extend our method to the one-stage NavSim Dauner et al. (2024) benchmark to further validate its robustness in end-to-end planning. Our perception module follows the Transfuser Chitta et al. (2022b) architecture, where multi-modal LiDAR and image features are fused into the BEV feature. In the baseline, ego query fuses the current BEV feature via cross-attention and directly decodes the future trajectory. Our implementation of OccDriver on NavSim currently incorporates dual-branch architecture and occupancy collision loss. Transformer-based dual-branch architecture iteratively decodes coarse trajectory, future occupancy and fine trajectory. Here, OccDriver predicts occupancy for both non-drivable areas and agents. Guided by occupancy prediction, occupancy collision loss regularizes the trajectory to avoid off-road deviations and collisions.

Table. 16 presents the comparison of end-to-end driving scores on the NavSim Test split. Under the evaluation of PDM score, OccDriver attains superior driving score over the baseline, demonstrating its robustness and generalization capability. The boost in the Progress (EP) metric is attributed to the prediction of future occupancy, which enhances the model's understanding of scene dynamics. The gains observed in Collision (NC) and drivable-area-compliance (DAC) metrics further validate the effectiveness of occupancy guidance. Occupancy guidance loss provides strong constraints on keeping away from occupied areas, promoting safer planning behavior.

## J  ADDITIONAL QUALITATIVE RESULTS

Fig. 7 illustrates the overlapping results of marginal and joint occupancy predictions to validate the effectiveness of contingency planning in modeling behavioral uncertainty. It showcases that marginal occupancy distribution not only captures the waiting behavior but also reflects the aggressive right turn possibility. Interacting with short-term marginal occupancy distribution avoids running into unexpected sudden situations, thus improving safety and comfort. Relevant quantitative results are presented in the ablation study.

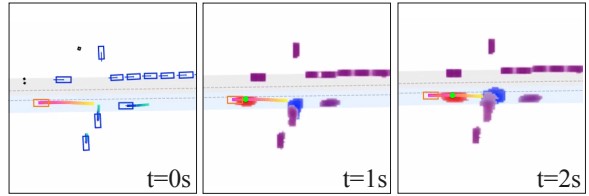

Figure 7: Visualization of contingency planning. Compared to joint occupancy (purple), marginal occupancy (blue) models potential behaviors.

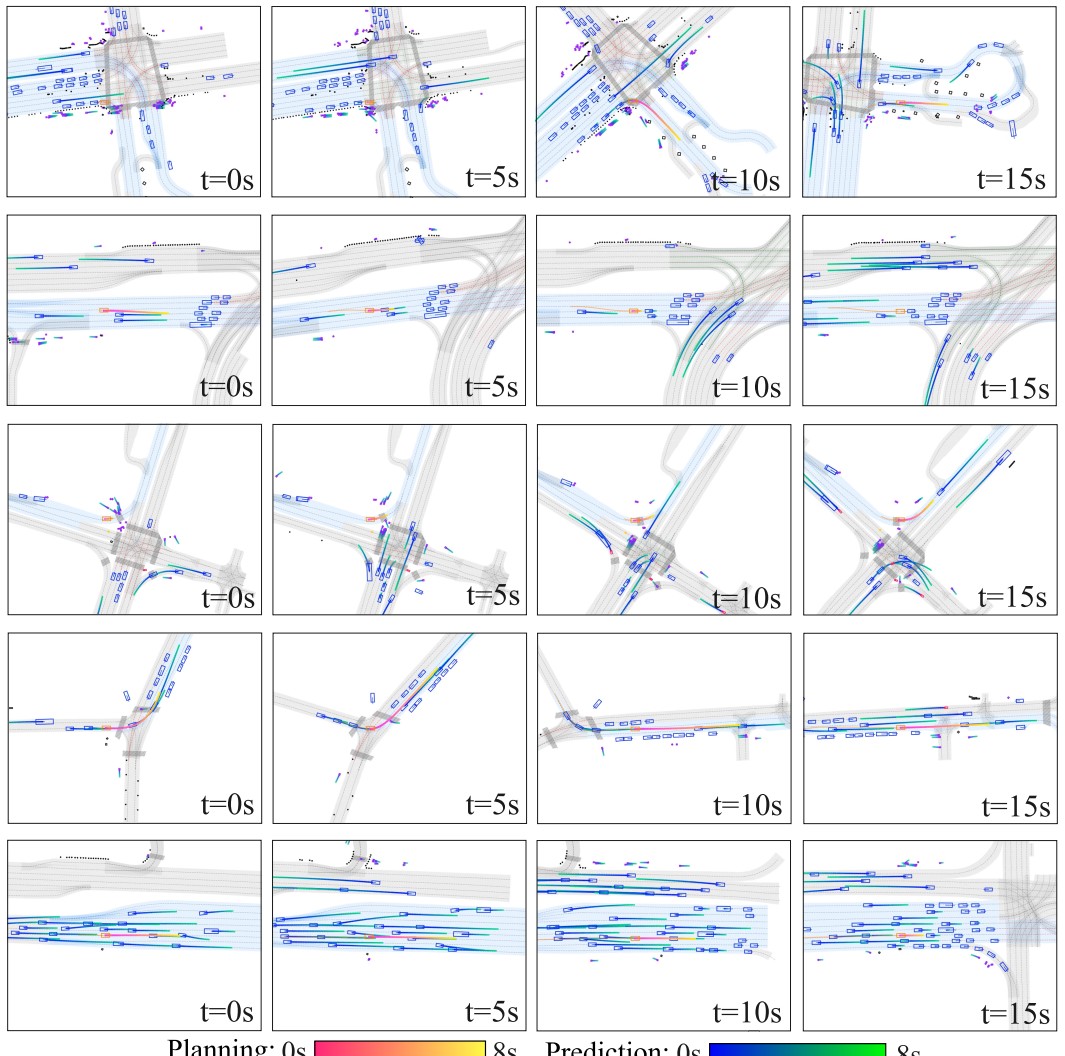

Figure 8: Visualization of closed-loop planning in nuPlan **Val14** benchmark.

Qualitative closed-loop Planning results on nuPlan **Val14** benchmark is provided in Fig. 8. Qualitative results of occupancy guidance is provided in Fig. 9.

## K    VIDEO RESULTS

We include several 15-second closed-loop planning videos on **Test14 − Hard** split in the supplementary material, which intuitively demonstrates OccDriver's closed-loop planning performance.

## L    CODE

We provide the core code of the iterative decoding structure, specialized loss functions and agent pruning mechanism in the supplementary material to facilitate reproducibility.

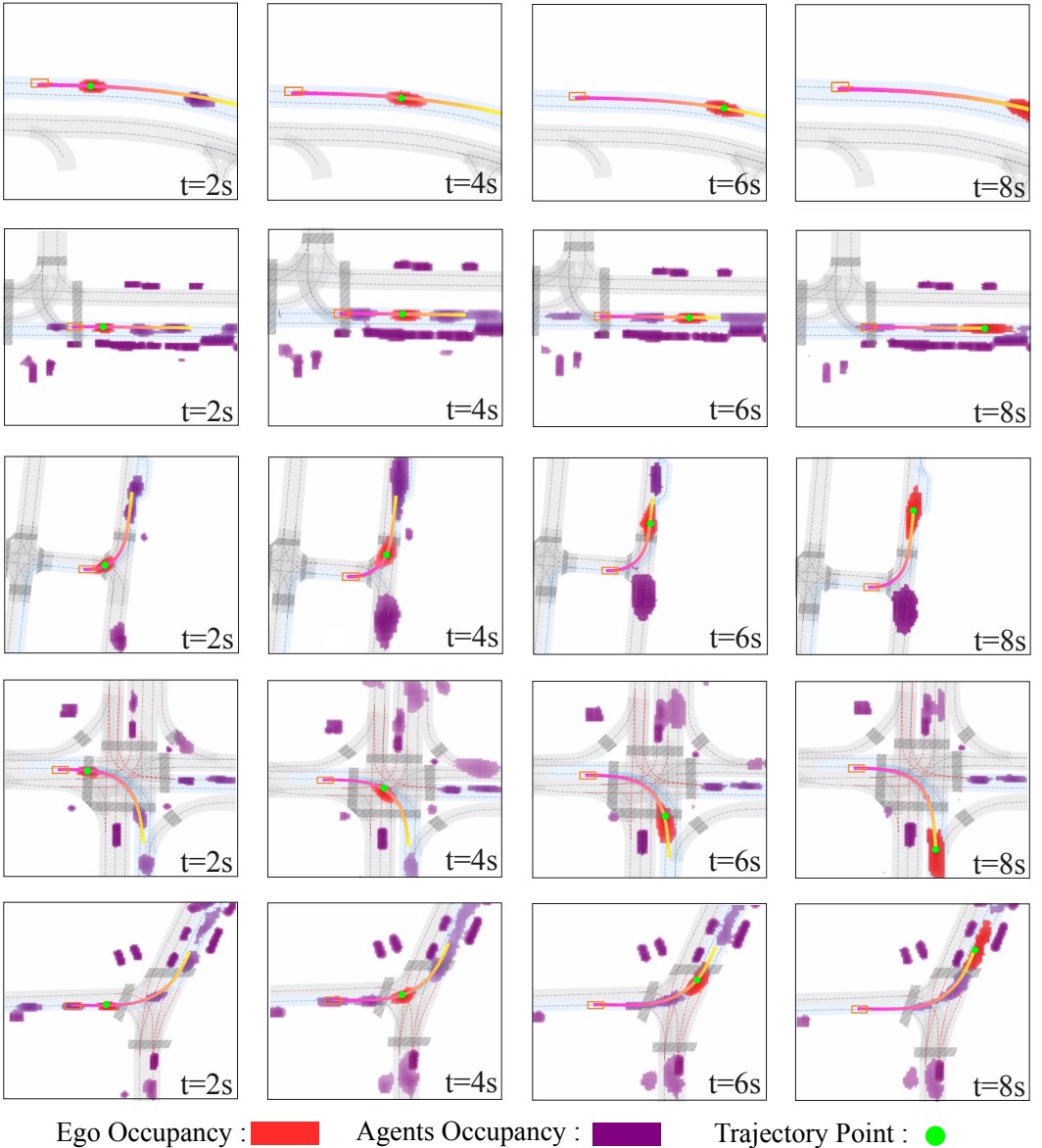

Figure 9: Visualization of closed-loop planning under guidance of future occupancy prediction.

