# OpenReview forum: "OccDriver: Future Occupancy Guided Dual-branch Trajectory Planner in Autonomous Driving"
_ICLR.cc/2026/Conference — ICLR 2026 Poster_

### Official Review · Reviewer_aAjV · 2025-10-29

**Soundness:** 3
**Presentation:** 3
**Contribution:** 3
**Rating:** 8
**Confidence:** 5

**Summary:**

This paper proposes OccDriver, a dual-branch transformer framework that integrates future occupancy prediction into trajectory planning for autonomous driving. The model consists of a vectorized branch, which generates coarse-to-fine multimodal trajectories, and rasterized branches, which predicts future occupancy and flow fields for both joint and marginal prediction. The rasterized branch acts as a world model, providing future scene evolution guidance to refine the ego trajectory. Several cross-modality consistency losses, including occupancy interference, trajectory-occupancy alignment, and trajectory-occupancy collision losses, are designed to couple the two branches effectively. The authors further introduce a contingency planning objective that considers both short-term marginal and long-term joint occupancy distributions to handle behavioral uncertainty. Evaluations on the nuPlan benchmark show that OccDriver achieves state-of-the-art performance on both Non-Reactive and Reactive closed-loop metrics, especially improving safety (collision, TTC) without sacrificing comfort or progress.

**Strengths:**

The proposed rasterized-to-vectorized pipeline elegantly combines probabilistic world-model reasoning with precise trajectory planning, bridging two dominant paradigms (rasterized and vectorized) in motion planning.

Incorporating marginal and joint future occupancy prediction as explicit supervisory signal and guidance mechanism through contingency planning, seems to be a quite novel idea

The experiments are thorough, covering both Val14 and Test14-Hard nuPlan benchmarks, with convincing performance gains in safety and closed-loop metrics. The ablation studies clearly demonstrate the contribution of each module.

**Weaknesses:**

The method adds nontrivial computation due to occupancy decoding and marginal distribution estimation. Although inference time is reported (≈23 ms), the training cost and scalability with agent count are not analyzed in depth.

The paper lacks a deeper analysis of why occupancy-based guidance ensures optimal or safer planning.

The fonts are small in current figures. Fig.2 a bit cluttered and difficult to follow. The implementation of contingency planning could also be summarized in a figure.

**Questions:**

1. Does marginal occupancy pruning significantly affect safety performance?

2. How are the short-term (Ts) and long-term (Tf) horizons chosen for contingency, and how robust is performance to these hyperparameters?

3. How does OccDriver conceptually differ from recent world-model planners (e.g., DriveDreamer) that also predict scene evolution before planning? Could OccDriver be extended to reinforcement-based fine-tuning?

---

> ### Author Response · Authors · 2025-11-19
>
> Reply for Weaknesses:
>
> 1. Thanks for your professional comment on training costs. The training cost breakdown of our model is reported in the table below. OP and MP denote occupancy prediction and marginal prediction, respectively. According to the table, the total computational costs of our model is relatively low. Occupancy decoding accounts for nearly half of the total computational overhead, due to the rasterized feature representation and additional losses. Marginal distribution prediction introduces minimal computational overhead, a result of its lightweight model architecture.
> | Base | OP | MP | Training Latency(ms) | Params(M) |
> |:---:|:------------------:|:----------------------:|:---------------------:|:---------:|
> | ✔ |  |  | 29.77 | 3.8 |
> | ✔ | ✔ |  | 60.61 | 7.4 |
> | ✔ | ✔ | ✔ | 69.24 | 7.9 |
>
>     In the second table, we analyse scalability of  the number of agents used in marginal prediction. A consistent increase in GPU memory allocation was observed as the agent count grows. Predicting marginal occupancy distributions for all agents incurs computationally prohibitive memory overhead. To address this issue, We leverage agent pruning to keep only interactive agents (typically fewer than 10), significantly enhancing scalability.
>     | Agent Num (bs=8) | GPU Memory (GB) |
>     |:---------:|:----------------:|
>     |     0     |      3.67        |
>     |     5     |      5.36        |
>     |    10     |      7.98        |
>     |    20     |      13.22       |
>
> 2. In our work, occupancy guidance yields benefits in two key aspects: information complementation and interaction modeling.
>
>     On one hand, our dual-branch framework fuses vectorized and rasterized feature representations, and the incorporation of scene-level occupancy information supplements individual features with global scene context and spatial dynamics.
>
>     On the other hand, we employ occupancy as explicit interaction modeling to guide safer planning. The probabilistic nature of occupancy captures behavioral uncertainty more effectively than ground-truth or trajectory regression. Marginal prediction and contingency planning further enhance the planner’s risk awareness.
>
> 3. Following your suggestions, we will revise the figures and update them in the final manuscript.

---

> ### Author Response · Authors · 2025-11-19
>
> Reply for Questions:
>
> 1. The ablation results of agent pruning is shown in table below. Because of GPU memory constraint, we select 8 nearest non-interactive agents, together with interactive agents (kept in agent pruning), for marginal occupancy prediction. The results indicate a minor drop in safety performance after agent pruning, validating the effectiveness of proposed pruning in keeping interaction-relevant agents. With agent pruning, our model attains a favorable balance between safety and progress.  More importantly, agent pruning substantially reduces the computational overhead of marginal prediction, thereby improving the model’s scalability.
> | Agent Prune | Collision |  TTC  | Progress |  NR-S  |
> |:-----------:|:---------:|:-----:|:--------:|:------:|
> |   without   | **0.951** | **0.915** | 0.866 | 0.840 |
> |     with    | 0.946 | 0.914 | **0.873** | **0.845** |
>
> 2. Following your thoughtful comment, we conduct separate ablation studies on the short-term (Ts) and long-term (Tf) horizons in contingency planning.
>
>     The first table below presents the ablation study on the long-term horizon Tf. We observe a consistent growth in Collision metric and driving scores as the horizon Tf increases, reaching peak at Tf = 6s. Driving performance starts to degrade when Tf extends to 8s. It suggests that long-horizon future occupancy captures scene dynamics and agents' long-term interactions, facilitating interaction-consistent planning. However, as the uncertainty of occupancy prediction accumulates over time, leveraging highly uncertain future occupancy as guidance introduces ambiguous or incorrect supervision signal, ultimately degrading planning performance.
>     |  Tf | Collision |  TTC   | NR-S | R-S |
>     | :-: | :--------: | :----: | :------: | :-----: |
>     | 2s  |   0.933    | 0.908  |  0.828   |  0.787  |
>     | 4s  |   0.943    | 0.916|  0.843   |  0.799  |
>     | 6s  | **0.946**  | 0.914  | **0.845** | **0.801** |
>     | 8s  |   0.931    | **0.917**  |  0.828   |  0.787  |
>
>     The ablation results for the short-term horizon Ts are reported in the second table below. The consistently increasing safety metrics, particularly TTC, indicate that marginal prediction effectively reprensents behavioral uncertainty, thereby enhancing the model's risk awareness. However, at Ts = 3s, excessive marginal uncertainty induces overly conservative planning, which degrades both progress and driving score.
>     |  Ts  | Collision |  TTC   |  NR-S  |  R-S   |
>     | :--: | :-------: | :----: | :----: | :----: |
>     |  1s  |   0.942   | 0.907  | 0.841  | 0.799  |
>     |  2s  |   0.946   | 0.914  | **0.845** | **0.801** |
>     |  3s  | **0.947** | **0.923** | 0.835  | 0.794  |
>
> 3. Generative world models focus on understanding future scene dynamics, in which interaction information is implicitly embedded.  However, OccDriver explicitly extracts and utilizes future interaction information to enhance planning safety. The exclusivity between the ego and agents' occupancy is leveraged to represent spatial interactions, and occupancy guidance losses distill interaction information into trajectory planning. Marginal prediction and contingency planning further improve planner’s risk awareness. The main difference between OccDriver and other world-model planners lies in its innovative utilization of future scene prediction as an interaction modeling paradigm.
>
>  4. Thank you for your enlightening question, which motivated us to consider the RL extension of our work. While occupancy prediction cannot serve as a closed-loop RL simulator like video- or 3DGS-based world models, the differentiable occupancy guidance losses can be modified into reward functions to punish collision-prone trajectories. And we will explore RL application for contingency planning, constructing risk-aware optimization objectives such as Conditioned Value at Risk (CVaR).

---

> ### Comment · Reviewer_aAjV · 2025-11-20
>
> Thanks for the detailed reply. I believe that using contingency planning to incorporate occupancy guidance is indeed a meaningful direction for strengthening closed-loop planning.
>
> In contrast to Rew #HoqV’s comments, I would argue that the contribution is not merely a simple or incremental “guidance” mechanism as in UniAD. Rather, the combination of contingency is novel.
>
> Also, from my expertise, I do not think NavSim is an appropriate benchmark to assess this aspect. NavSim is designed for end-to-end modeling and could not provide a clear evaluation of long-term closed-loop performance.

---

> > ### Author Response · Authors · 2025-11-20
> >
> > We sincerely appreciate your professional review and your recognition of our work. Please let us know if you have any further questions or points for discussion.

---

### Official Review · Reviewer_sotP · 2025-10-31

**Soundness:** 3
**Presentation:** 3
**Contribution:** 3
**Rating:** 6
**Confidence:** 3

**Summary:**

This paper presents OccDriver, a rasterized-to-vectorized framework for motion planning in autonomous driving. Motivated by the limitations of trajectory-based or occupancy-based approaches, it integrates a vectorized trajectory decoder with an occupancy world model that predicts future scene evolution conditioned on coarse trajectories. Cross-modal losses—including occupancy interference, trajectory–occupancy alignment, and collision penalties—enable explicit safety-aware guidance. Additionally, a contingency strategy based on short-term marginal and long-term joint occupancies enhances robustness under uncertainty.Evaluated on the nuPlan benchmark, OccDriver achieves SOTA closed-loop driving scores and superior safety metrics, demonstrating its effectiveness in reliable and interpretable planning.

**Strengths:**

- The paper proposes an innovative Dual-branch Planning Framework with a coarse-to-fine decoding mechanism: the coarse trajectory decodergenerates a preliminary behavioral framework, while the fine trajectory decoder optimizes trajectory with future scene information. This hierarchical design balances planning efficiency and accuracy, addressing the computational cost-rationality trade-off in traditional single-stage methods.
- Marginal prediction (MP) and contingency planning (CP) modules are introduced for emergency risk: MP captures individual agents’ short-term marginal occupancy, and CP generates emergency trajectories. Ablation experiments (Table 3) confirm both modules improve safety metrics and driving scores, verifying the risk-aware design’s practical value.

**Weaknesses:**

- The paper demonstrates the effectiveness of the proposed Dual-branch Planning Framework by presenting the performance of the fine trajectory that incorporates future scene information and occupancy guidance. However, it lacks direct quantitative and qualitative comparative analysis between the coarse trajectory and the fine trajectory.6.2 The paper only dissolved M=1, 3, and 6, without verifying
the monotonic performance growth of more M or identifying the optimal "peak" M. If there is a maximum peak M, there is still a lack of strict explanation for why a specific M (if optimal) performs the best.

**Questions:**

- Given that the fine trajectory decoding is conditioned on the latent variableQ_c(coarse trajectory feature), what is the necessity of explicitly decoding the coarse trajectoryY_c? Additionally, is there confusion regarding whether direct supervision is applied toY_c?
- Supplementing direct quantitative and qualitative comparisons between the coarse trajectory and the fine trajectory would more effectively demonstrate the validity of the "coarse-to-fine" framework proposed in this paper.
- The authors should conduct more comprehensive ablation studies on the number of modalities.

---

> ### Author Response · Authors · 2025-11-19
>
> Reply for Weaknesses:
>
> 1. The table below compares the planning metrics between coarse and fine trajectories.  The quantitative result shows that fine trajectories achieve improved safety metrics and driving scores over coarse trajectories. It demonstrates that the integration of future scene features, together with explicit occupancy guidance, provides positive information for trajectory planning, enabling refinement into safer trajectories.
>
>     In Appendix.H, we additionally provide the qualitative comparison on Test14-Hard set in Figure.6. Compared with collision-prone coarse trajectories, fine trajectories are evidently altered to avoid collisions and handle interactions effectively, validating the effectiveness of the coarse-to-fine architecture and occupancy guidance.
>     |    | Collisions | TTC | NR-S | R-S |
>     |:---:|:----------:|:-----:|:--------:|:-------:|
>     | Coarse Traj  | 0.931     | 0.903 | 0.836    | 0.790   |
>     | Fine Traj | **0.946**     | **0.914**| **0.845**   | **0.801**   |
>
> 2. Following your suggestion, we conduct a comprehensive ablation study on the number of modalities.  As presented in the table below, M=9 yields optimal driving scores(0.848 NR-S and  0.805 R-S). It indicates that M=9 is sufficient to cover the ego’s longitudinal behaviors. More modality enables finer-grained division of ego’s behavioral patterns, leading to an continuous growth of Progress metric. Both Collision and TTC degrade when M exceeds 9, which we attribute to overly redundant modality partitioning that reduces trajectory flexibility. Besides, the increased number of modalities reduces the training data allocated to each modality, eventually causing insufficient training and degraded trajectory quality.
> |  M  | Collision |  TTC   | Comfort | Progress | NR-S  | R-S  |
> | :-: | :-------: | :----: | :-----: | :------: | :---: | :--: |
> |  1  |   0.915   | 0.887  |  0.975  |  0.860   | 0.823 | 0.755 |
> |  3  |   0.931   | 0.904  | **0.978** |  0.869   | 0.835 | 0.784 |
> |  6  |   0.946   | **0.914** |  0.976  |  0.873   | 0.845 | 0.801 |
> |  9  | **0.951** | 0.911  |  0.972  |  0.876   | **0.848** | **0.805** |
> | 12  |   0.944   | 0.898  |  0.965  | **0.882** | 0.840 | 0.797 |
>
> Reply for Questions:
>
> 1. Thank you for your thoughtful question on the necessity of coarse trajectory. Inspired by the intermediate-layer supervision widely used in Transformer framework, we impose trajectory regression loss (Smooth L1 loss,  loss weight = 0.3) to coarse trajectory Y_c.  Direct trajectory supervision ensures that Q_c models meaningful trajectory information, easing the learning of  subsequent occupancy prediction and trajectory refinement, and stabilizing the overall training process. In particular, occupancy prediction requires reliable initial behaviors to guarantee high-quality decoding of scene evolution. Additionally, supervision on Y_c avoids having Q_c rely solely on deep-layer gradients, thereby alleviating vanishing-gradient issues and improving model robustness and interpretability.

---

> ### Author Response · Authors · 2025-11-26
>
> Dear reviewer:
>
> As the discussion phase is approaching its end, we are looking forward to your valuable feedback regarding whether the above clarifications and the added experiments have addressed your concerns and questions. We would be happy to address any additional points you may have during the remaining time of the discussion phase.
>
> Thank you for engaging with us in the discussion.

---

### Official Review · Reviewer_HoqV · 2025-11-01

**Soundness:** 2
**Presentation:** 2
**Contribution:** 1
**Rating:** 2
**Confidence:** 4

**Summary:**

This paper proposes OccDriver, a dual-branch rasterized-to-vectorized trajectory planner that integrates future occupancy predictions as guidance for trajectory generation in autonomous driving. The framework consists of a rasterized branch that predicts future scene evolution in occupancy space and a vectorized branch that plans ego trajectories conditioned on this predicted scene. The authors utilize several cross-branch loss functions and a contingency planning objective to enhance safety and robustness. Experiments on the nuPlan dataset demonstrate state-of-the-art closed-loop results compared with existing learning-based planners. While the paper is technically sound and well-engineered, its conceptual novelty is insufficient for a top-tier contribution (see Weaknesses).

**Strengths:**

1. The quantitative results on the nuPlan benchmark show competitive performance.
2. The ablation study is detailed and systematic.
3. The paper is readable, the methodology is well-documented, and architectural details are described thoroughly in the appendices.

**Weaknesses:**

1. My main concern is the novelty of this paper. The concept of using occupancy prediction to guide planning is not new. Even UniAD[1] in 2023 has already explored joint designs between occupancy forecasting and trajectory generation. It seems that the main difference here is to adopt a parallel dual-branch structure rather than a cascaded one, which feels like an architectural variation rather than a fundamentally new paradigm. The claimed “future occupancy guidance” and “dual-branch feature interaction” are incremental extensions of prior occupancy-assisted pipelines.
2. The related work section misses several closely aligned approaches (e.g., some end-to-end methods like World4Drive[2])
3. The evaluation relies on a single dataset, which limits claims about robustness and generalization. To substantiate the method’s effectiveness, it should include additional benchmarks such as NavSim[3].


[1] Hu, Yihan, et al. "Planning-oriented autonomous driving." Proceedings of the IEEE/CVF conference on computer vision and pattern recognition. 2023.
[2] Zheng, Yupeng, et al. "World4Drive: End-to-end autonomous driving via intention-aware physical latent world model." Proceedings of the IEEE/CVF International Conference on Computer Vision. 2025.
[3] Dauner, Daniel, et al. "Navsim: Data-driven non-reactive autonomous vehicle simulation and benchmarking." Advances in Neural Information Processing Systems 37 (2024): 28706-28719.

**Questions:**

Minor:
1. Discuss failure cases and qualitative scenarios where occupancy guidance meaningfully alters the decision.
2. No ablation isolating the dual-branch architecture itself (without losses) is clearly analyzed.

---

> ### Author Response · Authors · 2025-11-19
>
> Reply for Weaknesses:
>
> 1. We appreciate your thoughtful concern on novelty.  Occupancy predictions have indeed been employed in earlier work to assist planning. UniAD uses occupancy prediction only for trajectory post-processing and most methods integrate future scene information via feature fusion. However, highly compressed features struggle to preserve fine-grained spatial interaction information, limiting occupancy's utility in enhancing planning safety.
>
>     In contrast, the key innovation of OccDriver is exploiting occupancy prediction as explicit interaction modeling. Beyond dual-branch feature fusion, we dig the mutual exclusivity between ego and agent occupancy to model spatial interactions, and design occupancy guidance losses to distill occupancy interaction into trajectory planning.
>
>    To further enhance planning safety, we leverage the probabilistic distribution nature of occupancy prediction to represent agent behaviors’ uncertainty. Accordingly, we apply marginal prediction and contingency planning strategy in occupancy space for risk-sensitive planning. Such unprecedented interaction modeling in occupancy space enables OccDriver to achieve state-of-the-art safety performance.
>
> 2. We appreciate your suggestion to supplement related work. In Appendix.A.1, we add related work about world models in autonomous driving and world-model-guided end-to-end planning. We categorize world-model guidance into explicit trajectory constraints (as in Drive-WM and Drive-OccWorld) and implicit feature transfer (as in LatentDriver and World4Drive). LatentDriver unifies the environment’s next states and ego’s next action as a mixture distribution. World4Drive constructs an intention-aware latent world model to generate and evaluate multimodal trajectories. And we compare our model with other world-model-guided planners. OccDriver extends beyond predicting future scenario evolution by further exploiting interaction modeling in occupancy world and how this can benefit planning safety.
>
> 3. Following your suggestion, we extend our method to the one-stage NavSim benchmark to further validate its robustness. Our perception module follows the Transfuser architecture, where multi-modal LiDAR and image features are fused into the BEV feature. In the baseline, ego query fuses the current BEV feature via cross-attention and directly decodes the future trajectory.
>
>     Our implementation of OccDriver on NavSim currently incorporates dual-branch architecture and  occupancy collision loss. Transformer-based dual-branch architecture iteratively decodes coarse trajectory, future occupancy and fine trajectory. Here, OccDriver predicts occupancy for both non-drivable areas and agents. Guided by occupancy prediction, occupancy collision loss regularizes the trajectory to avoid off-road deviations and collisions.
>
>     The table below presents the comparison of end-to-end driving scores on the NavSim Test. OccDriver attains superior driving score over the baseline. The boost in the Progress (EP) metric is attributed to the prediction of future occupancy, which enhances the model’s understanding of scene dynamics. The gains observed in Collision (NC) and drivable-area-compliance (DAC) metrics further validate the effectiveness of occupancy guidance. Occupancy guidance loss provides strong constraints on keeping away from occupied areas, promoting safer planning behavior.
>
>     As closed-loop simulation environment is not supported, NavSim benchmark cannot evaluate long-term closed-loop planning performance as nuPlan does. However, under the evaluation of PDM score, OccDriver achieves safe and efficient end-to-end planning, demonstrating its robustness and generalization capability.
>     | | NC   | DAC  | TTC  | CF  | EP   | PDM-S |
>     |:------------:|:------:|:------:|:------:|:-----:|:------:|:-------:|
>     | UniAD      | 97.8 | 91.9 | 92.9 | 100 | 78.8 | 83.4  |
>     | Transfuser | 97.9 | 92.8 | 92.8 | 100 | 79.2 | 84.0  |
>     | PARA-Drive | 97.9 | 92.4 | 93.0 | 99.8| 79.3 | 84.0  |
>     | Baseline   | 98.1 | 95.7 | **94.8** | 100 | 82.6 | 87.7  |
>     | OccDriver  | **98.4** | **97.5** | 94.3 | **100** | **84.0** | **89.2** |

---

> ### Author Response · Authors · 2025-11-26
>
> Reply for Questions:
>
> 1. The table below presents the qualitative comparison of planning metrics between coarse and occupancy-guided fine trajectory.  Under occupancy guidance, fine trajectory outperforms coarse trajectory in both safety metrics and driving scores.  In Appendix.H, Figure.6 illustrates the qualitative comparison between coarse and fine trajectories on Test14-Hard set. In interactive scenarios, fine trajectories alter collision-prone behaviors into safe trajectories,  demonstrating the beneficial influence of occupancy guidance on interaction awareness and planning safety. Figure.9 further visualizes the alignment between output trajectory and occupancy prediction. It shows that ego's trajectory point is positioned within the ego’s high-occupancy region and steers away from agents’ high-occupancy region, validating the effectiveness of occupancy guidance.
> |  | Collisions | TTC | NR-S | R-S |
> |:---:|:----------:|:-----:|:--------:|:-------:|
> | Coarse Traj  | 0.931     | 0.903 | 0.836    | 0.790   |
> | Fine Traj | **0.946**     | **0.914**| **0.845**   | **0.801**   |
>
>
> 2. The table below presents the ablation results for dual-branch architecture. The dual-branch architecture achieves superior safety and progress metrics compared to the vectorized-only baseline, demonstrating the effectiveness of proposed dual-branch structure. We attribute the enhanced planning performance to incorporating both current and future occupancy features from the occupancy branch. Current occupancy feature supplements global spatial information lost in the vectorized representation. Future occupancy feature further extracts scene-evolution information to refine scene-consistent trajectories.
> |   | Collision |  TTC   | Comfort | Progress | NR-S  | R-S  |
> | :------- | :-------: | :----: | :-----: | :------: | :---: | :--: |
> | Vec branch  |   0.917   | 0.884  | **0.976** |  0.885   | 0.836 | 0.769 |
> | Dual branch | **0.933** | **0.905** |  0.966  | **0.893** | **0.859** | **0.787** |

---

> ### Author Response · Authors · 2025-11-30
>
> Dear reviewer:
>
> As the discussion phase is approaching its end, we would greatly appreciate your feedback on whether the above clarifications and the added experiments have addressed your concerns and questions. If there are any additional points or questions, please let us know. Your feedback is valuable to us, and we are committed to addressing further issues to improve our work.
>
> Thank you for your time and effort in reviewing our paper.

---

### Meta-Review · Area_Chair_E2HS · 2026-01-08

**Summary:**

This paper proposes OccDriver, a dual-branch transformer framework that integrates future occupancy prediction into trajectory planning for autonomous driving. Evaluations on the nuPlan benchmark show that OccDriver achieves state-of-the-art performance on both Non-Reactive and Reactive closed-loop metrics, especially improving safety (collision, TTC) without sacrificing comfort or progress.

**Reviewer Concerns:**

Reviewer concerns are summarized as follows:

- Novelty limitation and lack of related works.
- Evaluation on a single dataset (which is fine from AC's perspective).
- Lack of numeric analysis between the coarse vs fine trajectory. More detailed technical clarity is needed.
- Training cost / scalability should be discussed.

**Reviewer Scores:**

The review sores are 6, 2, 8. In general authors did a good job addressing all of the concerns.

Regarding the negative 2 score, the major concern centers around novelty (to incorporate occupancy guidance). AC read the paper, review and rebuttal, and aligns with the other reviewer that, indeed the combination of contingency is novel, even compared with UniAD work (Hu et al, 2023).

Please revise the paper based on the review comments.

---

### Decision · Program_Chairs · 2026-01-26

Accept (Poster)